# Diversion of Acetyl CoA to 3-Methylglutaconic Acid Caused by Discrete Inborn Errors of Metabolism

**DOI:** 10.3390/metabo12050377

**Published:** 2022-04-21

**Authors:** Dylan E. Jones, Elizabeth A. Jennings, Robert O. Ryan

**Affiliations:** Department of Biochemistry & Molecular Biology, University of Nevada Reno, Reno, NV 89557, USA; dylanjones@unr.edu (D.E.J.); elizabethjennings@nevada.unr.edu (E.A.J.)

**Keywords:** organic aciduria, leucine, acetyl CoA, sirtuin 4, 3-methylglutaconic acid, mitochondria, inborn errors of metabolism

## Abstract

A growing number of inborn errors of metabolism (IEM) have been identified that manifest 3-methylglutaconic (3MGC) aciduria as a phenotypic feature. In primary 3MGC aciduria, IEM-dependent deficiencies in leucine pathway enzymes prevent catabolism of *trans*-3MGC CoA. Consequently, this metabolite is converted to 3MGC acid and excreted in urine. In secondary 3MGC aciduria, however, no leucine metabolism pathway enzyme deficiencies exist. These IEMs affect mitochondrial membrane structure, electron transport chain function or ATP synthase subunits. As a result, acetyl CoA oxidation via the TCA cycle slows and acetyl CoA is diverted to *trans*-3MGC CoA, and then to 3MGC acid. Whereas the *trans* diastereomer of 3MGC CoA is the only biologically relevant diastereomer, the urine of affected subjects contains both *cis*- and *trans*-3MGC acids. Studies have revealed that *trans*-3MGC CoA is susceptible to isomerization to *cis*-3MGC CoA. Once formed, *cis*-3MGC CoA undergoes intramolecular cyclization, forming an anhydride that, upon hydrolysis, yields *cis*-3MGC acid. Alternatively, *cis*-3MGC anhydride can acylate protein lysine side chains. Once formed, *cis*-3MGCylated proteins can be deacylated by the NAD^+^-dependent enzyme, sirtuin 4. Taken together, the excretion of 3MGC acid in secondary 3MGC aciduria represents a barometer of defective mitochondrial function.

## 1. Introduction

Urine organic acid analysis serves as a vital tool for the diagnosis and treatment of inborn errors of metabolism (IEM) [1]. It is well established that discrete IEMs give rise to signature organic acids that are detectable by gas chromatography–mass spectrometry (GC-MS) [2]. For example, mutations in *GCDH*, encoding glutaryl CoA dehydrogenase, are causative for glutaric acidemia type I, wherein large amounts of glutaric acid are excreted in urine. Likewise, mutations in *PCC*, encoding propionyl CoA carboxylase, result in excretion of copious quantities of propionic acid. These signature organic acids arise from blockage of specific metabolic pathways. In the case of glutaric acidemia, lysine metabolism is disrupted by an IEM in *GCDH* such that the mutant enzyme is unable to oxidatively decarboxylate the pathway intermediate, glutaryl CoA, to crotonyl CoA and CO_2_. When this occurs, glutaryl CoA levels rise, CoA is cleaved and glutaric acid is excreted in urine. In a similar manner, mutations in *PCC* prevent the conversion of propionyl CoA to methylmalonyl CoA by the biotin containing enzyme, propionyl CoA carboxylase, leading to the excretion of propionic acid. As such, signature profiles of organic acids provide valuable diagnostic information and report on therapeutic efficacy. In other organic acidurias, however, the situation is more complex. As described below, 3-methylglutaconic (3MGC) aciduria not only occurs in deficiencies of leucine metabolic pathway enzymes, but also in a growing number of disparate IEMs that affect mitochondrial function. In this review, we describe a novel biochemical route to 3MGC acid from acetyl CoA and describe how a series of non-enzymatic chemical reactions affect pathway directionality and the end products formed. In addition, the breadth of IEMs that give rise to 3MGC aciduria as a distinguishing phenotypic feature is described.

## 2. Primary and Secondary 3MGC Aciduria 

Two classes of 3MGC aciduria have been identified, termed primary and secondary [3]. Primary 3MGC aciduria is caused by mutations in *HMGCL*, encoding 3-hydroxy-3-methylglutaryl (HMG) CoA lyase [4] or AUH, encoding 3MGC CoA hydratase (AUH) [5]. Both enzymes function in the leucine degradation pathway such that a deficiency in either one blocks complete catabolism of this amino acid. In the leucine degradation pathway (Figure 1A), AUH catalyzes hydration of trans-3MGC CoA to (S)-HMG CoA, which is then cleaved by HMG CoA lyase, generating acetoacetate and acetyl CoA. When IEMs lead to a deficiency in either of these enzymes, upstream pathway intermediates, including *trans*-3MGC CoA, accumulate. As with other organic acidurias, it has been proposed that, when trans-3MGC CoA and other pathway intermediates reach a threshold level, thioester hydrolysis yields the corresponding organic acids that appear in urine (Figure 1B) [6]. Dietary supplementation with (L)-leucine increases the amount of organic acid excreted in primary 3MGC aciduria, indicating no alternate catabolic route exists [7]. With regard to treatment of primary 3MGC aciduria, it is largely symptomatic, although dietary restriction of leucine intake may be beneficial.

By contrast, in secondary 3MGC aciduria, no leucine pathway enzyme deficiencies exist. Compared to primary 3MGC aciduria, the amount of 3MGC acid excreted is much lower in secondary 3MGC aciduria and the amounts excreted are unaffected by (L)-leucine supplementation. Additional differences include the lack of excretion of 3-hydroxyisovaleric (3HIV) acid and a lower ratio of 3MGC acid to 3-methylglutaric (3MG) acid in secondary 3MGC aciduria [6]. As described below, distinct proposals have been advanced to explain the metabolic origin of 3MGC acid in secondary 3MGC aciduria. 

## 3. Mevalonate Shunt Hypothesis and Secondary 3MGC Aciduria 

Barth Syndrome (BTHS) is a rare X-linked disorder that presents with secondary 3MGC aciduria, often in conjunction with hypocholesterolemia [8]. This observation led to the hypothesis that disruptions in sterol or isoprenoid metabolism cause mevalonic acid to enter a “mevalonate shunt” pathway [9,10]. According to this proposal, dimethylallyl pyrophosphate is dephosphorylated to the corresponding free alcohol in two steps. The alcohol is then oxidized to 3-methylcrotonic acid, which is activated by the addition of CoA to produce 3-methylcrotonyl CoA, an intermediate in the leucine degradation pathway [11]. Once cytosolic isoprenoids are redirected to mitochondria and the leucine degradation pathway, it is proposed that they are converted to 3MGC acid and excreted in urine. Since BTHS is caused by an IEM in the *TAFAZZIN* gene, encoding the cardiolipin transacylase, tafazzin, this hypothesis fails to address why *trans*-3MGC CoA would not be processed normally to (*S*)-HMG CoA and then to acetoacetate and acetyl CoA [12]. In a similar manner, the mevalonate shunt hypothesis does not provide a physiological rationale to explain why isoprenoid metabolites would be redirected from cytosol to mitochondria as a result of a mutation that specifically affects mitochondrial phospholipid metabolism. Furthermore, this hypothesis fails to explain why *trans*-3MGC CoA generated by the mevalonate shunt pathway would not proceed down the leucine degradation pathway via hydration to (*S*)-HMG CoA by AUH, nor does it explain the presence of *cis*-3MGC acid in secondary 3MGC acidurias. 

## 4. The Acetyl CoA Diversion Pathway

In 2014, Su and Ryan proposed that *trans*-3MGC CoA can be synthesized de novo from acetyl CoA via the “acetyl CoA diversion pathway” [12]. According to this hypothesis, specific IEMs that impair mitochondrial structure or function result in compromised energy metabolism. As the efficiency of aerobic respiration declines, reduced cofactors (i.e., NADH and FADH_2_) produced during oxidative metabolism of fuel molecules are unable to efficiently donate electrons to the electron transport chain (ETC). As the concentration of reduced cofactors increases in the mitochondrial matrix, they serve as end product inhibitors of enzymes that generate NADH or FADH_2_, including the TCA cycle enzymes, isocitrate dehydrogenase [13] and α-ketoglutarate dehydrogenase [14]. This, in turn, inhibits TCA cycle activity and oxidation of acetyl CoA. When this occurs in muscle or heart tissue, some portion of the acetyl CoA pool is hypothesized to be diverted to *trans*-3MGC CoA in three steps (Figure 2) catalyzed by acetoacetyl CoA thiolase (T2), HMG CoA synthase 2 and AUH, respectively [6,12]. In this pathway, AUH-mediated dehydration of (*S*)-HMG CoA proceeds in the opposite direction of the leucine catabolism pathway (i.e., hydration of *trans*-3MGC CoA). Once formed, *trans*-3MGC CoA is unable to proceed further up the leucine degradation pathway because the reaction catalyzed by the next enzyme, 3-methylcrotonyl CoA carboxylase (3MCCCase), is irreversible [15]. Consistent with this, unlike primary 3MGC aciduria, no increase in the amount of 3HIV acid occurs in secondary 3MGC aciduria. In primary 3MGC aciduria, 3HIV acid is derived from 3-methylcrotonyl CoA, following hydration and thioester cleavage. However, because 3-methylcrotonyl CoA cannot be produced via the acetyl CoA diversion pathway, this organic acid is absent in secondary 3MGC aciduria [7].

Another distinction between primary and secondary 3MGC acidurias relates to the ratio of *trans*-3MGC acid to *cis*-3MGC acid recovered in urine. In primary 3MGC aciduria, a 2:1 *cis*:*trans* diastereomeric ratio is observed while, in cases of secondary 3MGC aciduria, the ratio is closer to 1:1 [7]. Until recently, unanswered questions pertaining to both primary and secondary 3MGC aciduria included: (1) an explanation for the presence of significant amounts of *cis*-3MGC acid in urine of subjects with 3MGC aciduria and (2) characterization of factors that drive AUH-mediated dehydration of (*S*)-HMG CoA in the acetyl CoA diversion pathway, despite the fact that the equilibrium constant of this enzyme strongly favors the reverse reaction (i.e., hydration) [16].

## 5. Isomerization of *trans*-3MGC CoA

Support for the acetyl CoA diversion pathway was obtained following recognition that *trans*-3MGC CoA is susceptible to non-enzymatic isomerization to *cis*-3MGC CoA [17]. In the leucine degradation pathway, as well as the acetyl CoA diversion pathway, *trans*-3MGC CoA is exclusively formed. In leucine catabolism, as depicted in Figure 1, *trans*-3MGC CoA is formed by 3MCCCase-mediated carboxylation of 3-methylcrotonyl CoA while, in the acetyl CoA diversion pathway, (*S*)-HMG CoA is dehydrated to *trans*-3MGC CoA by AUH [6]. Since *trans*-3MGC CoA is considered to be the precursor of 3MGC acid, it may be expected that urine samples would exclusively contain the *trans* diastereomer. However, ^1^H-NMR studies revealed that urine of patients with primary 3MGC aciduria contains an ~2:1 ratio of *cis*- to *trans*-3MGC acid [18,19]. A mixture of both diastereomers was also observed upon GC-MS analysis of urine samples from patients with BTHS [8]. Importantly, when pure *trans*-3MGC acid was analyzed by GC-MS, two chromatographic peaks, corresponding to *cis*- and *trans*-3MGC acid, respectively, were observed [20]. In studies designed to investigate this unexpected result, Jones et al. [21] reported that commercially available *trans*-3MGC acid undergoes non-enzymatic isomerization to *cis*-3MGC acid as a function of time and temperature. ^1^H-NMR spectroscopy of *trans*- and *cis*-3MGC acid standards at 25 °C revealed no evidence of isomerization, suggesting that elevated temperatures reached during GC promote isomerization of 3MGC acid. Whereas both diastereomers undergo isomerization as a function of increasing temperature and incubation time, under identical incubation conditions, *trans*-3MGC acid isomerizes to a greater extent. This result suggests that *cis*-3MGC is stabilized by the formation of an intramolecular hydrogen bond (Figure 3). Importantly, however, when *trans*-3MGC acid was incubated at 37 °C, the rate of isomerization proceeded too slowly (in the order of days) to account for the ratio of *cis*- and *trans*-3MGC acid found in urine. Based on these results and computational analysis, a chemical mechanism describing the isomerization reaction has been proposed [21]. As discussed below, *trans*-3MGC CoA is much more reactive than *trans*-3MGC acid, and this enhanced reactivity has important biological consequences.

## 6. A Non-Enzymatic Reaction Sequence Converts *trans*-3MGC CoA to *cis*-3MGC Acid 

As *trans*-3MGC CoA is produced by the dehydration of (*S*)-HMG CoA in the acetyl CoA diversion pathway, it isomerizes to *cis*-3MGC CoA. Unlike *trans*-3MGC CoA, *cis*-3MGC CoA can undergo intramolecular cyclization to form *cis*-3MGC anhydride and free CoA [17]. The resulting cyclic anhydride is reactive and can be hydrolyzed to yield the dead-end organic acid, *cis*-3MGC acid, or it can undergo nucleophilic attack by protein lysine side chain amino groups to form a covalent adduct (Figure 4). Based on this non-enzymatic reaction sequence, a potential route from *trans*-3MGC CoA to *cis*-3MGC acid exists [22]. Not only does this reaction sequence provide an explanation for the presence of *cis*-3MGC acid in urine of subjects with primary and secondary 3MGC aciduria, it also explains how AUH’s equilibrium constant [16], which strongly favors hydration over dehydration, is overcome. This reaction sequence proceeds because non-enzymatic isomerization of *trans*-3MGC CoA to *cis*-3MGC CoA induces further AUH-mediated dehydration of (*S*)-HMG CoA in order to re-establish the equilibrium substrate/product ratio. By removing one AUH reaction product, in this case *trans*-3MGC CoA, (*S*)-HMG CoA will continue to be consumed. In essence, isomerization of *trans*-3MGC CoA to *cis*-3MGC CoA, and its further conversion to a cyclic anhydride, functions as a “sink” that culminates in production of *cis*-3MGC acid. Thus, a key structural difference between *cis*- and *trans*-3MGC CoA is the inability of the latter to form a cyclic anhydride. Wagner et al. [23] reported that various terminally carboxylated short-chain acyl CoAs, including glutaryl CoA, 3-methylglutaryl (3MG) CoA and (*S*)-HMG CoA, spontaneously undergo intramolecular cyclization to form the corresponding cyclic anhydride, with loss of CoA. Once formed, these cyclic anhydrides have two possible fates, hydrolysis to the carboxylic acid or acylation of protein lysine side chain amino groups. The hydrolysis of *cis*-3MGC anhydride exclusively generates *cis*-3MGC acid, thereby providing a molecular explanation for the occurrence of *cis*-3MGC acid in urine of subjects with 3MGC aciduria. In the case of protein 3MGCylation, in vitro studies [22] revealed that, although *cis*-3MGC anhydride displays a preference for hydrolytic cleavage versus acylation, considerable protein 3MGCylation also occurs. The fact that Young et al. [17] detected 3MGCylated bovine serum albumin (BSA) following incubation of *trans*-3MGC CoA with BSA provides compelling evidence in support of the non-enzymatic phase of the acetyl CoA diversion pathway. In genetically modified mouse models that manifest 3MGC aciduria, consistent with the site of 3MGC CoA formation, mitochondrial proteins were found to be 3MGCylated [24].

## 7. Sirtuin 4-Mediated Deacylation 

Once formed, 3MGCylated proteins can be deacylated through the action of sirtuin 4 (SIRT4) [24], producing *cis*-3MGC acid as a product. SIRT4, which is found specifically in mitochondria, is a member of a family of NAD^+^-dependent deacylase enzymes that catalyze cleavage of amide bonds formed between acyl moieties and protein lysine side chain amino groups. The reaction requires NAD^+^ as a co-substrate, and yields the following products: deacylated protein, nicotinamide and 2′-*O*-acyl-phosphoribose (Figure 5). The latter product subsequently undergoes non-enzymatic hydrolysis to yield ADP-ribose and the corresponding organic acid [25]. Studies with SIRT4-knockout mice [25] revealed the role of this enzyme in removal of 3MGC- adducts from substrate proteins. Although it has been reported [26] that *trans*-3MGC- is the acyl moiety bound to protein lysine side chains, evidence now exists that it is actually the *cis*-diastereomer of 3MGC- that forms an amide linkage with substrate proteins [22].

## 8. A Model IEM Associated with Secondary 3MGC Aciduria

As mentioned above, BTHS is a rare, X-linked IEM that manifests several characteristic phenotypic features including dilated cardiomyopathy, skeletal myopathy, growth delay, neutropenia and 3MGC aciduria [27]. BTHS affects approximately 300 young males worldwide, although recent studies indicate this disorder may be greatly underdiagnosed [28]. BTHS is caused by mutations in the *TAFAZZIN* gene which encodes the phospholipid transacylase, tafazzin, that is responsible for acyl chain remodeling of cardiolipin (CL) [27,29]. CL is a unique phospholipid found primarily in mitochondrial cristae, highly curved, cylinder-shaped membranes that house ETC complexes [30]. While the fatty acyl composition of CL varies among tissues, highly oxidative tissues, including heart and skeletal muscle, contain ~90% tetralinoleoyl CL [27]. Mutations in *TAFAZZIN* that result in a deficiency in tafazzin activity manifest (i) decreased levels of tetralinoleoyl CL, (ii) greater CL acyl chain heterogeneity and (iii) increased amounts of monolyso CL [31,32]. One outcome of these changes is an increase in the ratio of monolyso CL to tetralinoleoyl CL [33]. Consistent with CL’s role in maintenance of cristae membrane structural integrity, BTHS mitochondria display morphological and functional abnormalities, including cristae membrane adhesion, collapse of the inner cristae space and partial uncoupling of oxidative phosphorylation [34,35,36]. These changes to cristae membrane structure adversely affect ETC function and reduce ATP production capacity. Just as dilated cardiomyopathy in BTHS arises from insufficient ATP production [37], as ETC efficiency declines, NADH and FADH_2_ cofactor levels rise in the matrix space. Subsequent inhibition of TCA cycle activity prevents acetyl CoA entry, resulting in its diversion to 3MGC acid via the enzyme-mediated, and non-enzymatic, stages of the acetyl CoA diversion pathway [4]. 

## 9. Other IEMs Associated with 3MGC Aciduria

Since the first reports of secondary 3MGC aciduria, numerous additional IEMs have been described that manifest this phenotypic feature. Wortmann et al. [38] described various IEMs and delineated several factors that distinguish primary from secondary 3MGC aciduria. Subsequently, Su and Ryan [12] reported on a broader range of IEMs associated with this phenotype. More recently, Jones et al. [4] cataloged 20 distinct IEMs in which secondary 3MGC aciduria is observed. These include components of ATP synthase [39], genes that encode mitochondrial lipid metabolic enzymes [40,41], a mitochondrial protein disaggregase [42] as well as metabolic enzymes of the TCA cycle [43]. Thus, it is apparent that a broad range of enzyme deficiencies can give rise to secondary 3MGC aciduria. The observation that, in every case, the responsible IEM affects some aspect of mitochondrial energy metabolism is consistent with the acetyl CoA diversion pathway model.

## 10. Conclusions

Whereas studies to date have uncovered novel concepts related to 3MGC acid biosynthesis, a number of questions remain, including potential tissue specific differences in the production of 3MGC acid. With regard to the detection and quantitation of 3MGC acid in biological fluids, it is conceivable that a recently described polyclonal antibody directed against 3MGC acid [17] will permit the development of a rapid immunoassay for 3MGC acid. This would circumvent delays normally associated with classical organic acid analysis by GC-MS and could be adapted to a quantitative ELISA format. If so, it would then be possible to measure excreted 3MGC acid levels in real time and correlate these with physiological processes including exercise, drug therapy or nutritional status. By correlating measurable physiological parameters with 3MGC acid excretion, disease status and/or mitochondrial function can be precisely monitored in real time. Likewise, another key area to explore is the extent to which protein 3MGCylation in these disorders impacts biological processes, including enzyme activity or metabolic pathway flux [24]. Studies to determine the efficiency with which SIRT4 removes 3MGC moieties, and analysis of the spectrum of proteins that are 3MGCylated, will provide new insight into this rapidly advancing field.

## Figures and Tables

**Figure 1 metabolites-12-00377-f001:**
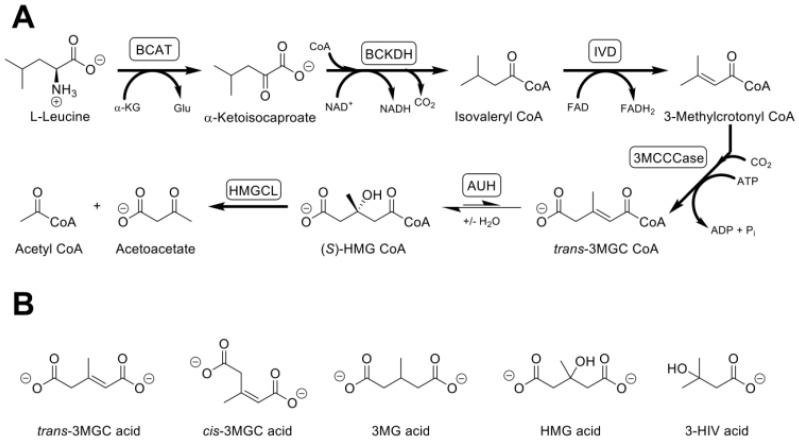
The leucine catabolism pathway. (Panel (**A**)) L-leucine is initially converted to *α*-ketoisocaproate in a reaction catalyzed by branched chain aminotransferase (BCAT) which employs *α*-ketoglutarate (*α*-KG) as amino group acceptor, yielding glutamate (Glu). *α*-Ketoisocaproate is then oxidatively decarboxylated by branched chain *α*-keto acid dehydrogenase (BCKDH) to form isovaleryl CoA. Subsequently, isovaleryl CoA dehydrogenase (IVD) oxidizes isovaleryl CoA to 3-methylcrotonyl CoA. This metabolite is then carboxylated to *trans*-3MGC CoA by 3-methylcrotonyl CoA carboxylase (3MCCCase). *trans*-3MGC CoA is then dehydrated by AUH to form (*S*)-HMG CoA which is cleaved by HMG CoA lyase (HMGCL), forming acetoacetate and acetyl CoA. (Panel (**B**)) Inborn errors of metabolism that lead to a deficiency in either AUH or HMGCL prevent complete catabolism of L-leucine, resulting in the excretion of organic acids, including *trans*-3MGC acid, *cis*-3MGC acid, 3-methylglutaric (3MG) acid, HMG acid (only in HMGCL deficiency) and 3-hydroxyisovaleric (3-HIV) acid.

**Figure 2 metabolites-12-00377-f002:**
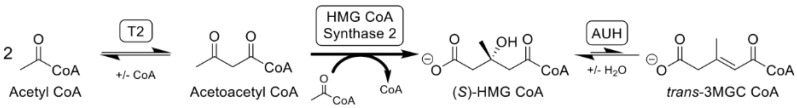
The acetyl CoA diversion pathway in secondary 3MGC acidurias. When IEMs occur in genes that affect mitochondrial membrane structure, ETC function or ATPase subunits, acetyl CoA entry to the TCA cycle slows. When this occurs, acetyl CoA is diverted to *trans*-3MGC CoA in three enzyme-mediated steps. Initially, acetoacetyl CoA thiolase (T2) catalyzes a Claisen condensation between two acetyl CoA, yielding acetoacetyl CoA and free CoA. Following this, HMG CoA synthase 2 catalyzes a second Claisen condensation between acetoacetyl CoA and acetyl CoA, yielding (*S*)-HMG CoA and free CoA. This intermediate is then dehydrated to *trans*-3MGC CoA by the leucine catabolism pathway enzyme, AUH.

**Figure 3 metabolites-12-00377-f003:**
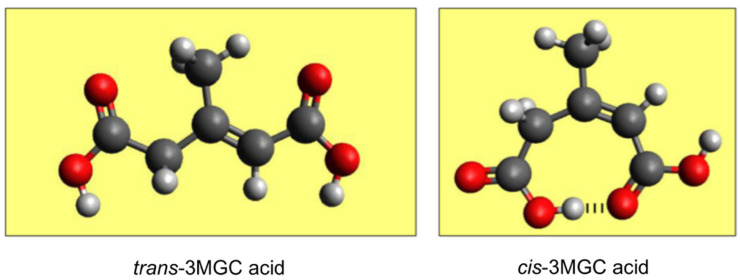
Model depicting enhanced stability of *cis*-3MGC acid versus *trans*-3MGC acid. Energy minimized molecular model of *trans*-3MGC acid and *cis*-3MGC acid. The potential for intramolecular hydrogen bonding between carboxylate groups in *cis*-3MGC acid (see dashed lines), but not *trans*-3MGC acid, provides an explanation for the apparent higher stability of the *cis* diastereomer.

**Figure 4 metabolites-12-00377-f004:**
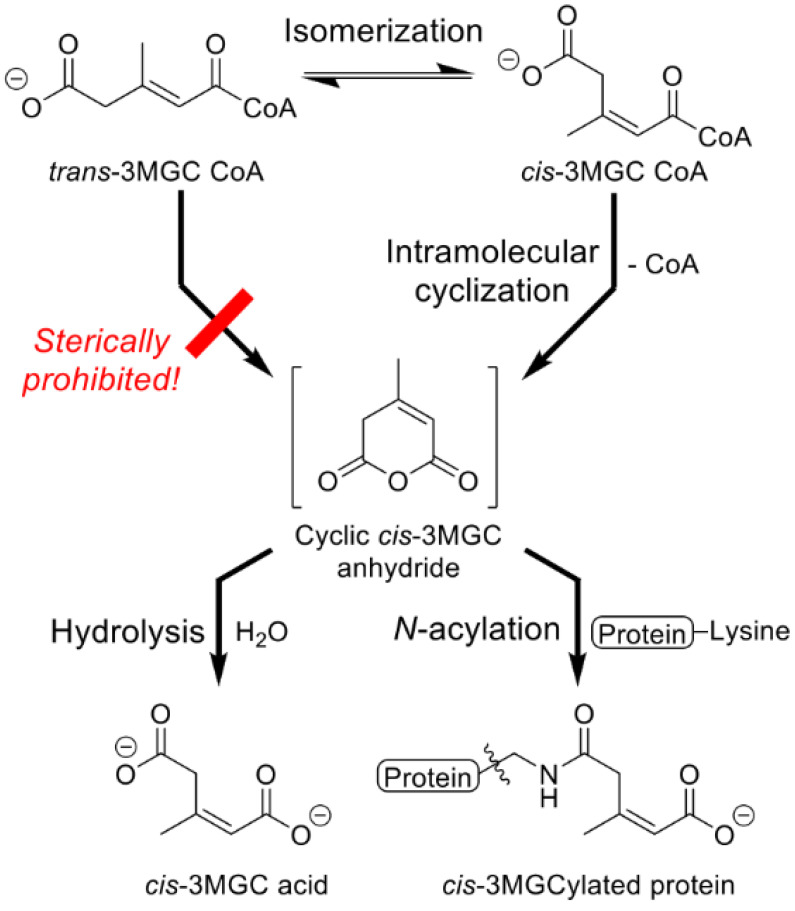
Non-enzymatic reactions downstream of *trans*-3MGC CoA. Once formed, trans-3MGC CoA can readily isomerize to *cis*-3MGC CoA. A key difference between these diastereomers is the ability of *cis*-3MGC CoA, but not *trans*-3MGC CoA, to undergo intramolecular cyclization, yielding *cis*-3MGC anhydride and free CoA. Once formed, *cis*-3MGC anhydride has two possible fates, including hydrolysis of the anhydride to form the organic acid, *cis*-3MGC acid or *N*-acylation of protein lysine side chains.

**Figure 5 metabolites-12-00377-f005:**
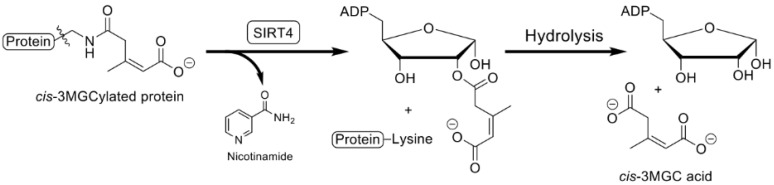
SIRT4-mediated deacylation of 3MGCylated proteins. As 3MGCylated proteins are generated via the non-enzymatic phase of the acetyl CoA diversion pathway, they become substrates for SIRT4 deacylase activity. This reaction, which employs NAD^+^ as a co-substrate, initially yields nicotinamide and 2-*O*-*cis*-3MGC ADP ribose as products. Subsequent non-enzymatic hydrolysis of 2-*O*-*cis*-3MGC ADP ribose yields the products, ADP ribose and *cis*-3MGC acid.

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
