# Peer review of "Diversion of Acetyl CoA to 3-Methylglutaconic Acid Caused by Discrete Inborn Errors of Metabolism"

_metabolites, 2022, doi:10.3390/metabo12050377_

Round 1
Reviewer 1 Report
Dylan E. Jones et al summarize inborn errors of metabolism (IEM) that lead to primary or secondary 3-methylglutaconic (3MGC) aciduria. They investigate scenarios linking mitochondrial disfunction to accumulation of this metabolite. The manuscript is well written and the topic is really interesting and could provide insights at the service of multiple fields, from basic science to the clinic. I strongly recommend publication in this journal after minor corrections and clarifications.
- Please specify how thioester hydrolysis which generates correspondent organic acids is regulated.
- It would be interesting to indicate current treatments for primary 3MGC aciduria.
- When authors state that the mevalonate shunt hypothesis does not provide complete explanation for the phenotype observed, authors should provide and speculate further about reasons and possibilities for mechanisms.
- Authors should consider if isomerization at 37C in urine and kidneys beforehand is possible at all: they could provide a rough estimate of time from production to excretion of the metabolites studied. Maybe if it is a few days the temperature-driven isomerization could be justified?
- Are there any other evidence for protein acylation besides in vitro BSA? Is there data available on modified proteins in patients? Is anything known about the spectrum of proteins that are 3MGCylated? Moreover, authors should clarify if the residues modified are limited to just Lys.
- Authors should be more specific about paper in ref 25. They should provide details on which proteins have been found to be modified. More details also about paper in ref 22 should be provided, specifically regarding how it was discovered that the cis-diastereomer is responsible for the modification.
- Since Sirt 4 is mitochondrial, authors should comment more about that. Are only mitochondrial proteins subjected to the discussed modification then?
- An important point involves the fact that authors could be more clear about subcellular localization of the reactions they describe and how transport of these metabolites in /out of mitochondria vs cytosol and outside the cell is regulated. This would also help recollecting with some statements about Sirt4 involvement.
- Authors could comment about ADP ribose. Is this metabolite contributing to any phenotype in patients with errors?
- It would be important to draw in a figure the model proposed in paragraph 8 with mitochondria, enzymes, cristae etc.
- Authors should clarify how antibodies that can recognize metabolites are made.
Reviewer 2 Report
This review by Jones et al. provides up to date information on the inborn errors of metabolism (IEM) that give rise to 3-methylglutaconic (3MGC) aciduria and the metabolism of primary and secondary 3MGC aciduria. In addition to the classical pathways, they describe a novel biochemical route to 3MGC acid from acetyl CoA and how a series of non-enzymatic chemical reactions may affect the directionality of this pathway and the end products formed. Overall, this is a well written manuscript that covers the subject area in adequate depth and represents a timely and comprehensive review of the metabolism of 3MGC aciduria in classical IEM with important focus on the controversial metabolic source of 3MGC in these IEMs.
Major points:
- For the reader it would be nice for the authors to include a Table listing the known IEMs in which primary and secondary 3MGC occurs.
- Lines 100, 242, 247 change TAZ to the newly appropriate gene name TAFAZZIN
Author Response
General Comment:
This review by Jones et al. provides up to date information on the inborn errors of metabolism (IEM) that give rise to 3-methylglutaconic (3MGC) aciduria and the metabolism of primary and secondary 3MGC aciduria. In addition to the classical pathways, they describe a novel biochemical route to 3MGC acid from acetyl CoA and how a series of non-enzymatic chemical reactions may affect the directionality of this pathway and the end products formed. Overall, this is a well written manuscript that covers the subject area in adequate depth and represents a timely and comprehensive review of the metabolism of 3MGC aciduria in classical IEM with important focus on the controversial metabolic source of 3MGC in these IEMs.
Response: We thank the reviewer for this comment
Major points:
- For the reader it would be nice for the authors to include a Table listing the known IEMs in which primary and secondary 3MGC occurs.
Response: A table containing this information has been reported (see text reference [4], which is cited on lines 51 and 278).
2. Lines 100, 242, 244 change TAZ to the newly appropriate gene name TAFAZZIN
Response: We thank the reviewer for this comment. The text has been edited to use the term TAFAZZIN (see line 95, 254 and 259).